# Utilization of Cottonseed Meal Protein Hydrolysate by Crustaceans: Insights on Growth Performance, Protein Turnover, and Metabolism in Chinese Mitten Crab *Eriocheir sinensis*

**DOI:** 10.3390/ani13233631

**Published:** 2023-11-23

**Authors:** Chaofan He, Wenbin Liu, Ling Zhang, Weiliang Chen, Zishang Liu, Xiangyu Qian, Xiangfei Li

**Affiliations:** Key Laboratory of Aquatic Nutrition and Feed Science of Jiangsu Province, College of Animal Science and Technology, Nanjing Agricultural University, No. 1 Weigang Road, Nanjing 210095, China; 2020205025@stu.njau.edu.cn (C.H.);

**Keywords:** aquaculture nutrition, diet supplement, plant protein, gene expression, crustacean culture

## Abstract

**Simple Summary:**

Advancing the utilization of plant proteins is of great significance for aquaculture. Here, the effects of cottonseed meal protein hydrolysate (CPH) on the growth performance and protein metabolism of Chinese mitten crab were investigated. CPH enhanced protein deposition through the activation of the IGF-1/Akt/TOR and IGF-1/Akt/FoxO1 pathways, thereby promoting growth performance. This facilitates the development of effective nutritional interventions to promote the growth of crustaceans.

**Abstract:**

Plant protein hydrolysates could enhance the growth performance and diet utilization of aquaculture species. The mechanisms underlying their beneficial effects, however, remain unclear. The purpose of this study was to appraise the effects of cottonseed meal protein hydrolysate (CPH) supplementation on the growth performance, amino acid profiles, and protein turnover and metabolism of *Eriocheir sinensis*. A total of 240 crabs (initial weight: 37.32 ± 0.38 g) were randomly assigned to six groups, and fed six iso-protein feeds supplemented with varying levels of 0% (the control group), 0.2%, 0.4%, 0.8%, 1.6% and 3.2% of CPH. These diets were continuously fed to the crabs for 12 weeks. The findings demonstrated that, compared with the control group, adding 0.4–0.8% CPH to the diet significantly increased the specific growth rate, nitrogen retention efficiency, hepatopancreas index, body crude protein content, hepatopancreas alanine aminotransferase and glutamine synthetase activities, hemolymph total protein content, the hepatopancreas transcription of S6 kinase-poly-peptide 1, and the hepatopancreas protein levels of insulin-like growth factor-1 (IGF-1), protein kinase B (Akt), and target of rapamycin (TOR) of crabs. In contrast, when the dose of dietary CPH reached 3.2%, the forkhead box O1 (FoxO1) protein expression showed a significant decrease compared with the control group. In addition, CPH supplementation also increased the amount of amino acids and free amino acids in hepatopancreas and hemolymph, respectively. Together, these findings demonstrated that dietary supplementation of 0.4–0.8% CPH could activate the IGF-1/Akt/TOR pathway of *E. sinensis*, thereby improving growth performance, protein synthesis, and utilization. For cost considerations, the recommended dietary dose of CPH for *E. sinensis* is 0.8%. Due to the above benefits, CPH has the potential to be used as a growth promoter for other aquatic animals, especially crustaceans.

## 1. Introduction

Chinese mitten crab (*Eriocheir sinensis*) is widely distributed in the north and south shores of China with an average production of 770,482 t in the last three years [1,2,3]. Because of its rich flavor and tender meat, *E. sinensis* is commonly consumed by Chinese residents, making China one of the world’s largest markets for this species. In the traditional production of *E. sinensis*, biological baits such as trash fish have been widely adopted, and this inevitably leads to environmental pollution and disease outbreaks [4]. This is not in compliance with the concept of green aquaculture in China. In contrast, artificial feeds have the advantages of comprehensive nutrition, controlled quality and safety, easy storage, and low cost. With the implementation of environmental protection policies, the Chinese government has gradually restricted the use of trash fish in *E. sinensis* aquaculture by promoting artificial feed. However, *E. sinensis* fed artificial feed generally exhibited slower growth and longer culture cycles than the ones fed with trash fish. In addition, consumers like to buy *E. sinensis* during the Chinese traditional festivals such as the Mid-Autumn Festival and National Day, during which period the price of this species reaches the highest level. However, at this time, *E. sinensis* fed with artificial feed are not yet fully mature, and cannot be sold to the market in a timely manner, thus seriously affecting the economic benefits. Therefore, finding effective nutritional interventions to promote the growth and maturation of *E. sinensis* is of great significance to popularize the use of artificial feed in practical aquaculture. 

With the development of modern food processing technologies, plant protein hydrolysates (PPHs) have attracted the attention of aquaculture researchers due to their high nutritional values, antimicrobial functions, antioxidant capacities, and sensory properties [5]. Generally, PPH could be obtained by treating traditional plant proteins with enzymes under specific pH and temperature conditions. They have a low content of anti-nutritional factors, and are enriched with amino acids, small peptides, and soluble proteins compared with the traditional plant proteins, endowing them with broad application prospects in aquafeed. Cottonseed meal protein hydrolysate (CPH) had the best effect on growth and immunity among other types including rapeseed, soybean, and peanut meals [6] and was also able to stimulate appetite and organic matter accumulation in *E. sinensis* [4]. In addition, other studies have indicated that dietary CPH supplementation at appropriate amounts could promote the growth performance of fish [7]. Furthermore, replacing a small portion of fishmeal with CPH can reduce the feed cost without compromising the growth rates of turtle and fish [8,9]. These results all suggest that CPH has the potential to be used as a growth promoter in aquafeed. However, the potential mechanisms underlying these beneficial effects are still poorly unveiled. Whether CPH functions through the regulation of protein metabolism and turnover is still unknown. 

As the main effector of growth hormone (GH), insulin-like growth factor-1 (IGF-1) plays a crucial function in regulating protein metabolism in the body [10]. Generally, IGF-1 could activate protein kinase B (Akt) which is a serine/threonine kinase [11]. Then, Akt targets two major downstream effectors (namely the target of rapamycin (TOR) and forkhead box O1 (FoxO1)) to further regulate protein synthesis and degradation, respectively [12]. Specifically, TOR could activate the S6 kinase-poly-peptide 1 (S6K1) and 4E blinding protein 1 (4EBP-1), thereby promoting protein synthesis and the body growth of animals [13,14]. Meanwhile, Akt can also reduce protein degradation by blocking the transcriptional activities of FoxO family members by sequestering them, thereby reducing protein degradation. Among them, FoxO1 is the most highly phosphorylated protein, which is able to regulate protein catabolism by regulating lysosomal and proteasomal degradation [15]. To date, the correlations between PPH and the IGF-1/Akt/TOR and IGF-1/Akt/FoxO1 pathways are still barely understood in aquatic animals. Recently, a previous study showed that dietary protein variations resulted from the enzymatic digestion of plant proteins, potentially affecting the TOR pathway in fish [16]. This suggested that PPH could regulate the protein synthesis and growth rate of aquatic animals by targeting TOR. However, the potential mechanisms are still barely elucidated.

Taking this into consideration, the objective of this research was to evaluate the impacts of CPH on the growth performance, body amino acid profiles, and protein turnover and metabolism in *E. sinensis*. The results obtained could advance the development of effective nutritional interventions to conquer the growth retardation and delayed marketability of *E. sinensis* fed artificial feed, thereby promoting the application of cottonseed meal and its by-products in aquaculture.

## 2. Materials and Methods

### 2.1. Animal Ethics

All animal procedures were approved by the Animal Care and Use Committee at Nanjing Agricultural University (ethical code: SYXK (Su) 2011 0036).

### 2.2. CPH and Diets

CPH was prepared using the techniques outlined previously [8]. Briefly, cottonseed meal (crude protein content: 54%) purchased from Xinjiang (China) was crushed into a fine powder. Then, a 100-mesh sieve was used to filter the powder. The materials that passed the sieve were thoroughly mixed with a neutral protease (AS1.398, Jiangsu Hipore Feed Co., Taizhou, Jiangsu, China). Then, a 5 h reaction was conducted (40 °C, pH 7.0, protease/substrate ratio 2500 IU/g). The enzymatic reaction was suspended by heating inactivation. After cooling, the solution was poured out, and the supernatant was concentrated with a rotary evaporator (evaporation temperature 65–70 °C), and was poured back and dried at 60 °C. Then, the dry supernatant powder and the dry precipitate were thoroughly mixed, through which the CPH used in this test was obtained. The amino acid profiles of CPH are as follows (g/kg): asparagine 49.39, serine 23.84, glutamic acid 108.98, glycine 22.45, alanine 22.29, cysteine 7.17, proline 19.9, threonine 17.68, valine 23.21, methionine 7.87, isoleucine 16.27, leucine 31.14, phenylalanine 29.73, lysine 22.13, histidine 14.94, and arginine 61.80. Then, graded levels (0, 0.2%, 0.4%, 0.8%, 1.6%, and 3.2%) of CPH were supplemented to the basal diet at the expense of cottonseed meal. The relative molecular mass distribution of the peptides in CPH was detailed previously [7]. 

Table 1 displays the formulations and approximate compositions of the experimental feeds. Table 2 displays the amount of amino acids in each experimental diet. Sources of protein included fish meal, soybean meal, blood meal, peanut meal, rapeseed meal, cottonseed meal, and peanut meal. The chosen lipid sources were fish oil and soybean oil, while the chosen carbohydrate source was α-starch. All feed ingredients were thoroughly mixed after grinding into fine powder. Then, the fat sources were added and blended to uniformity. Later, water was added (35% of raw material weight), and it was thoroughly blended with other materials. The feed was extruded using a single-screw meat grinder (die diameter 2.0 mm). Then, feed was air-dried, cut into a suitable length (1.8 cm), and kept at −20 °C.

### 2.3. Crab Management

Experimental crabs were obtained from the Shrimp and Crab Hatchery of Dongtai (Hipore Co., Ltd., Yancheng, China). Then, crabs were briefly cultivated in several cement pools for one week to adapt to the environment. After acclimatization, a total of 240 crabs (average weight: 37.32 ± 0.38 g, mean ± SD) were randomly distributed into 24 cement pools (1.0 × 1.0 × 0.8 m, L:W:H). Each pool (1.0 ×1.0 × 0.8 m, L:W:H) held ten crabs. Then, each diet was randomly assigned to four pools of crabs. The crabs were fed twice per day for 12 weeks. Clean water was drawn from an earthen pond to the cement pools, each of which had its own outlet valve and oxygenation pipeline. The water was changed every three days. In addition, aeration was provided when necessary. Water temperature, dissolved oxygen levels, pH, and ammonia were maintained at 24–28 °C, 5 mg/L, 8.0–8.5, and <0.05 mg/L, respectively, during the experimental period.

### 2.4. Sample Collection and Analysis

#### 2.4.1. Sample Collection

Crabs were starved for 24 h at the termination of the feeding experiment. Then, they were numbered and weighed to determine the growth performance parameters following anesthesia. Subsequently, three crabs were chosen at random within each pool with the hemolymph extracted from the base of the paraeiopod using 1 mL sterile syringes. The samples were mixed with the anticoagulant (100 mmol/L glucose, 26 mmol/L citrate, 30 mmol/L citric acid, 450 mmol/L NaCl, 10 mmol/L EDTA, and pH = 7.2) at 1:1 followed by centrifugation (900× *g* for 20 min at 4 °C). The supernatant was collected and stored at −20 °C for later use. Then, the hepatopancreases of these three crabs were quickly separated, weighed, and stored at −80 °C.

#### 2.4.2. Growth and Proximate Composition Analysis

The micro-Kjeldahl method was used to determine the crude protein content using an Auto Kjeldahl System (FOSS KT260, Hillerød, Denmark). Using the Soxtec System HT6 (Tecator, Haganas, Sweden), crude lipid was extracted by solvent extraction. Crude ash content was determined by burning at 550 °C for 4–6 h. To determine the moisture content, samples were dried at 105 °C to a constant weight. Crude fiber content was determined using an automatic fiber analyzer (ANKOM A2000i, ANKOM, Macedon, NY, USA). The growth performance data in this study was calculated as follows:

Specific growth rate (SGR, %/day) = (LnW_t_ − LnW_0_) × 100/T; 

Average body weight (ABW, g) = (W_0_ + W_t_)/2; 

Hepatopancreas index (HSI, %) = (hepatopancreas weight (g) × 100)/body weight (g);

Condition factor (CF) = (body weight (g) × 100)/total body length(cm)^3^; 

Protein efficiency ratio (PER, %) = wet weight gain (g)/total protein fed (g);

Nitrogen retention efficiency (NRE, %) = [(W_t_ × C_t_) − (W_0_ × C_0_)] × 100/(C_N_ × feed intake);

Mortality rate (MR, %) = 100 − (N_F_/N_I_) × 100;

W_t_ and W_0_ are the final and initial body weights of crab; T is the culture period in days; C_t_ and C_0_ are the final and initial contents in the whole body of crab, respectively; C_N_ is the nitrogen in diets; N_F_ and N_I_ are the final and initial number of crab.

#### 2.4.3. Enzyme Activity Assay

Aspartate aminotransferase (GOT) and alanine aminotransferase (GPT) activities in the hepatopancreas and hemolymph were both measured by using the commercial kits (number C010-1-1 and C009-2-1) manufactured by Nanjing Jiancheng Biological Technology Co., Ltd., (Nangjing, China). Briefly, the OD values of GOT and GPT were measured at 520 nm with a spectrophotometer (UV-1800, Shimadzu, Japan). Then, the GOT and GPT activities were calculated based on the standard curves. The hepatopancreas glutamine synthetase (GS) activity was also measured using a commercial kit (number A047-1-1) produced by Nanjing Jiancheng Biological Technology Co., LTD. (Nangjing, China). The activity of GS is expressed as per micromole of γ-glutamyl hydroxamate formed in per milligram of soluble protein per h. As specified previously [17], the hepatopancreatic calpain (CAPN) activity was assessed. The hepatopancreatic cathepsin L (catL) activity was evaluated in accordance with a previous study [18]. 

#### 2.4.4. Hemolymph Protein Metabolite Determination 

Coomassie brilliant blue staining was used to measure the hemolymph total protein (TP) content. Generally, protein molecules contain the -NH_3_^+^ groups, which can bind to anions in the Coomassie brilliant blue dye. This can turn the solution blue, and the protein content can be calculated by measuring the absorbance. In addition, an enzymatic (urease) colorimetric technique was used to determine the amount of urea nitrogen (UN) in the hemolymph. Briefly, urease was used to degrade urea into ammonia, which is later reacted with glutamate dehydrogenase to produce a fluorescent indicator. Then, the UN content can be calculated by measuring the absorbance.

#### 2.4.5. Amino Acid Analysis

For the analysis of total amino acid content, 0.2 g of the tested sample was placed into a 15 mL ampoule bottle containing 6 N hydrochloric acid (HCl) and was hydrolyzed for 24 h at 110 °C. After that, HCl was eliminated using a nitrogen-blowing instrument (Shanghai Jingfu Instrument Co., Ltd., Shanghai, China). The products obtained in the previous step were then re-dissolved with a loading buffer (0.1 N HCl), and were filtered via a 0.22 filter tip through an amino acid assay liner tube. Then, the amino acid content was analyzed using an amino acid standard mixing solution (Type H, Wako Pure Chemical Industries Ltd., Tokyo, Japan) for a fully automatic amino acid analyzer (L-8900, Hitachi High-Technologies, Inc., Tokyo, Japan). 

To evaluate the amount of free amino acids in hemolymph, samples were mixed with 10% sulfosalicylic acid, and were left to settle the protein at 4 °C for 10 min. Then, they were subjected to a 45 min centrifugation at 6500× *g*. The supernatant was filtered into the amino acid assay liner tube with a 0.22 um aqueous filter tip, and was subsequently analyzed using a fully automated amino acid analyzer (LA8080; Hitachi High-Technologies, Inc., Tokyo, Japan), and individual amino acids were determined by comparison with the standard (013-08391, Wako Pure Chemical Industries Ltd., Tokyo, Japan).

#### 2.4.6. Gene Expression Analysis

RNA extraction: A total of 0.1 g of hepatopancreas sample was taken and put into the EP tube with 1 mL of Trizol reagent (Sigma, Ronkonkoma, NY, USA), and then they were homogenized thoroughly at 4 °C. Then, the tube was placed in the ice box for 5 min, 200 μL of chloroform was added, and the container was shaken violently for 15 s before centrifugation at 12,000× *g* for 10 min. A total of 400 μL of supernatant was added to isopropanol at a ratio of 1:1, then they were mixed thoroughly at 25 °C for 10 min, and the same centrifugation was performed again. After aspirating the supernatant, 500 μL of ethanol at 75% by volume was added, mixed thoroughly, and spun for 5 min at 12,000× *g*. The precipitate was dried for 10 min at room temperature after the supernatant was removed. Next, 80 μL of diethyl pyrocarbonate (DEPC, Merck, Germany) water was added to lyse RNA. The quality and concentration of RNA were then determined using a Bioanalyzer (Agilent 2100 bioanalyzer, Agilent Technologies, Inc., Santa Clara, CA, USA) and a Nanodrop (Nanodrop 2000TM, Thermo Scientific, Wilmington, DE, USA), respectively. The RNA was diluted to the same concentration for reverse transcription.

RNA reverse transcription: After the RNA extraction, using a reverse transcription kit (PrimeScripte RT Master Mix, Takara, Kusatsu, Japan), cDNA was created. The reactions were performed in two steps: the first step at 42 °C for 40 min; and the second step at 90 °C for 2 min, followed by storage at 4 °C. After a 10-fold dilution, real-time fluorescence quantitative PCR (RT-qPCR) was carried out.

Quantitative real-time PCR: The reaction system of RT-qPCR was conducted by using 10 μL 2×ChamQ Universal SYBR qPCR Master Mix (Vazyme, Nanjing, China), 2 μL of cDNA template, 7.2 μL of DEPC water, and 0.4 μL of the forward and reverse primers, respectively. The PCR procedures included 95 °C for 30 s, and then 95 °C for 10 s, and 60 °C for 28 s for 40 cycles. The melting curve step was 95 °C to 60 °C for 15 s, and 60 °C for 1 min. Then, the temperature was raised to 95 ℃ at a rate of 1.6 °C per second for 15 s. At the end of the reaction, the specificity of the PCR product was determined by the melting curve.

Primer design and relative expression calculation: The Primer Premier 5.0 tool was used to create the primers for S6 kinase-polypeptide 1 (*s6k1*), 4E-binding protein 1 (*4ebp1*), and the *s27* (ubiquitin/ribosomal S27 fusion protein), and *β-actin*, with *s27* and *β-actin* serving as the internal reference genes, as previously employed [19,20]. The sequences of the primers, which were created by Shanghai Generay Biotech Co., Ltd. (Shanghai, China), are listed in Table 3. The gene amplification efficiency was measured and only primers with an amplification efficiency of >90% were used. The 2^−∆∆CT^ approach was used to assess the relative gene expression using the TaKaRa SYBR^®^
*Premix Ex Taq*^TM^ II kit (Takara, Kusatsu, Japan).

#### 2.4.7. Western Blot Analysis

Using a Tenbroeck tissue grinder (Kimble Chase, Vineland, NJ, USA) and a RIPA lysis buffer (50 mM Tris-HCl, 150 mM NaCl, 0.5% NP-40, 0.1% SDS and 1 mM EDTA), proteins were extracted from the hepatopancreas. The BCA protein assay kit was used to measure protein concentrations (number PA115, Tiangen, Beijing, China). Proteins were heat-denatured, and then electrophoresed using SDS-PAGE with a sample volume of 20 μg per lane. When the concentrated gel layer was penetrated by the bromophenol blue, the voltage was changed from 80 to 120 V using an electrophoresis system (Mini—PROTEAN Tetra System, Bio-Rad, Hercules, CA, USA). 

The proteins were transferred to a 0.45 um polyvinylidene fluoride (PVDF) membrane when the power was cut off. When the bromophenol blue moved to the lower border of the separation gel (Millipore, Burlington, MA, USA), the 5% skim milk powder closing solution was used to submerge the PVDF membranes, which were then placed on a shaker and slowly shaken. After 1 h of closure at 25 °C, the primary antibody was applied, the closed PVDF membranes were taken out, and were then incubated at 4 °C overnight. Following that, the membranes and a secondary antibody coupled to horseradish peroxidase were incubated at 25 °C for 2 h. Primary antibodies were insulin-like growth factor-1 (Igf-1, #AB182408, Abcam, Shanghai, China), protein kinase B, (Akt, #AB8805, Abcam, China), target of rapamycin (Tor, #AB134903, Abcam, China), forkhead box O1 (FoxO1, #AB52857, Abcam, China), and glyceraldehyde-3-phosphate dehydrogenase (Gapdh, #60004-1-Ig, Abcam, China). Finally, the bands were treated with the ECL reagent (GoodHere, Hangzhou, China) to visualize the bands. Using the ImageJ program, the intensity of the target bands was measured (U.S. National Institutes of Health, Bethesda, MD, USA).

### 2.5. Statistical Analysis

One-way ANOVA was performed to analyze the data using the SPSS (25.0) software program after checking the homogeneity (via the Shapiro–Wilk test) and the normality (via Levene’s test). Then, Tukey’s multiple range test was adopted to rank the means if there were significant differences among different groups. Meanwhile, an orthogonal polynomial comparison was also used to determine the type of significance (namely linear and quadratic) [23]. All percentage data were subjected to an inverse sinusoidal transformation prior to analysis [24]. *p* ≤ 0.05 was regarded as statistically significant, and all experimental data were presented as the S.E.M. (standard error of the mean) of four repetitions.

## 3. Results

### 3.1. Growth, Nutrient Utilization, and Body Composition

The dietary supplementation of CPH exerted no discernible influence (*p* > 0.05) on condition factor (CF), mortality rate (MR), and crude fat content (Table 4 and Table 5). However, the specific growth rate (SGR) and nitrogen retention efficiency (NRE) both increased significantly when CPH was added up to 0.8% (*p* < 0.01), but exerted no statistical difference with further elevated CPH levels (*p* > 0.05). The average body weight (ABW), protein efficiency ratio (PER), and crude protein content all increased significantly when CPH was added up to 1.6% (*p* < 0.05), but exerted little variation thereafter (*p* > 0.05). However, the reverse was true for moisture content. The hepatopancreas index (HSI) increased significantly when CPH was added up to 0.4% (*p* < 0.01) and then plateaued.

### 3.2. Amino Acid Profiles in Hemolymph and Hepatopancreas

Compared with the control group, the hemolymph Iso, Lys, His, and total amino acid contents in the CPH1.6 group were considerably higher (*p* < 0.05), while the CPH3.2 group had substantially higher hemolymph Thr, Val, and total essential amino acid levels (*p* < 0.05) (Table 6). The hepatopancreatic Asp content of the CPH0.4 group was considerably higher than that of the control group (*p* < 0.05), while the CPH0.8 group showed substantially higher contents of His and Cys (*p* < 0.05) (Table 7). In addition, compared to the control group, the CPH1.6 group displayed considerably higher levels of Thr, Ser, and Tyr (*p* < 0.05), and the CPH3.2 group showed substantially higher contents of Val, Met, Iso, Phe, Lys, Glu, total essential amino acids, total non-essential amino acids, and total amino acids (*p* < 0.05).

### 3.3. Biochemical Parameters in Hemolymph and Hepatopancreas

Dietary supplementation of CPH exerted no discernible impact on the aspartate aminotransferase (GOT), calpain (CAPN), and cathepsin L (catL) activities in the hepatopancreas, and the GOT and alanine aminotransferase (GPT) activities as well as the urea nitrogen (UN) content in the hemolymph (Figure 1 and Figure 2) (*p* > 0.05). However, the GPT activity in the hepatopancreas substantially increased, when CPH was added up to 0.2%, and also exerted a significant increase with further elevated dietary CPH levels (*p* < 0.05). With rising dietary CPH levels, there was a considerable rise in hemolymphatic total protein (TP) content (*p* < 0.05). Hepatopancreatic glutamine synthetase (GS) activity increased significantly when CPH was added up to 1.6% (*p* < 0.05), and showed no significance compared to that of the CPH3.2 group (*p* > 0.05).

### 3.4. Relative Expressions of s6k1 and 4ebp-1 in the Hepatopancreas

Compared to the control group, *s6k1* expression was significantly elevated when CPH was added at 0.8 (*p* < 0.05) (Figure 3), while *4ebp-1* expression showed the opposite trend (*p* < 0.05). 

### 3.5. Hepatopancreas Protein Expressions of the IGF-1/Akt Pathway-Related Proteins

The protein expressions of Igf-1, Akt, and Tor all increased significantly when CPH was added up to 0.8% compared to the control group (*p* < 0.05) (Figure 4). However, the protein expression of Foxo1 decreased significantly when the addition of CPH reached 1.6% (*p* < 0.05).

## 4. Discussion

In the current research, dietary supplementation with CPH remarkably enhanced the SGR, ABW, HSI, PER, NRE, and crude protein content, whereas the moisture content showed the opposite result. These results indicated that CPH supplemented at appropriate doses might improve the growth performance and protein utilization of *E. sinensis* and promote the accumulation of organic matter. According to a previous study, CPH is 125.00% higher in soluble protein, 59.38% higher in amino acids (74–180 Da), and 605.26% higher in small peptides (180–1983 Da) compared with cottonseed meal [7]. All of these enhancements could contribute to the promoted growth rate and protein utilization of *E. sinensis*. The increased PER and NRE are indicative of the reduced catabolism of protein, leaving more protein for cellular synthesis and development [25], as is further supported by the increases in body protein content. For the metabolism and storage of nutrients in *E. sinensis*, the hepatopancreas is a crucial organ, and its physiological condition is closely related to protein turnover and metabolism [26]. Accordingly, the enhanced protein deposition may partly contribute to the rise in HSI. 

The primary organ for protein digestion, absorption, and metabolism in *E. sinensis* is the hepatopancreas. In this research, the addition of CPH remarkably increased hepatopancreatic GPT activity, while no significance was noted in hemolymph GOT and GPT activities. This indicated that CPH supplemented at appropriate doses could benefit the protein metabolism of *E. sinensis* without damaging hepatopancreatic function. This was validated by the observation that enhanced amino acid metabolism is generally accompanied by a parallel increase in the activities of transaminases. This is consistent with a previous finding using hydrolyzed salmon pectoral fin protein to alleviate inflammatory damage in rats [27]. Supportively, the addition of CPH also markedly increased the hepatopancreatic activity of GS, which is one of the key enzymes for ammonia assimilation. Generally, increased GS activity is indicative of an enhancement in protein synthesis and nitrogen deposition [28], since GS is able to use glutamate as a precursor to consume NH_4_^+^ and ATP for the synthesis of glutamine. Glutamine is the most prevalent amino acid in blood, and a crucial building block for the creation of proteins and pyrimidine–purine nucleotides [29], thereby maintaining the balance of amino acid metabolism, and regulating the efficiency of protein synthesis [30]. Furthermore, it is noteworthy that CPH treatment significantly increased the hemolymph total protein content, suggesting enhanced protein absorption from diets. Generally, when an organism is under nutrient-rich conditions, the protein absorption capacity is elevated, thereby leading to increased TP concentration [31].

There are lots of nutrients stored in the hepatopancreas (especially proteins and lipids) of crustaceans in response to the nutritional needs of many physiological activities such as molting, gonad development, and starvation [26]. Additionally, the hepatopancreas has also been shown to be an essential organ for amino acid accumulation and protein synthesis [32,33]. In the current research, CPH treatment enhanced the concentrations of nearly every amino acid in the hepatopancreas of *E. sinensis* except for Leu, Arg, Gly, and Ala, indicating that CPH could promote amino acid storage and protein synthesis. Supportively, dietary supplementation of CPH also increased the concentrations of Val, Iso, total essential amino acids, and total amino acids in the hemolymph. This further reinforced the fact that CPH could stimulate amino acid absorption and protein synthesis in *E. sinensis*. This was supported by the evidence that the levels of free amino acids in hemolymph are indicative of amino acid absorption and protein synthesis in crustaceans [34]. According to a previous study [7], this result may be attributed to the 59.38% increase in the amino acid (74–180 Da) contents of CPH compared with the cottonseed meal. In addition, the increased amino acid contents in both the hepatopancreas and hemolymph may also help explain the increased body protein content in the CPH-treated groups. 

The TOR signaling pathway is closely involved in nutritional metabolism, especially in amino acid metabolism [35,36]. To unveil the molecular processes underlying the beneficial impacts of CPH on protein metabolism, the TOR signaling pathway was investigated at both transcriptional and translational levels. In the current research, the supplementation of CPH in feed enhanced the protein expressions of Igf-1, Akt, and Tor as well as the transcription of *s6k1* in the hepatopancreas, but decreased the protein expression of FoxO1 and the transcription of *4ebp-1*. This further indicated that CPH supplemented at appropriate doses could increase protein synthesis and inhibit protein catabolism in crustaceans, thus promoting protein deposition in the body. Supportively, the IGF-1/Akt/TOR axis and the IGF-1/Akt/FoxO1 axis are the main regulatory pathways of protein synthesis and catabolism, respectively [14,37]. IGF-1 is an essential growth factor with its level largely influenced by nutritional status [38]. Generally, the IGF-1 level increases to promote energy storage and protein synthesis when the body has sufficient nutritional conditions [39]. Accordingly, it can be speculated that dietary supplementation with CPH increases the level of IGF-1, which can consequently activate Akt through phosphorylation. The activated AKT in turn inactivates both tuberous sclerosis complex 1 and 2 to activate TOR [40,41], which then stimulates protein synthesis by phosphorylating S6K1 and inhibiting 4ebp-1 expression [42,43], from yeast to humans [44,45]. Meanwhile, the IGF-1 mediated activation of Akt could inhibit FoxO1 activation by phosphorylating several sites and anchoring them in the cytoplasm [46,47], thereby reducing protein degradation by controlling glycolipid catabolism and mitochondrial metabolism [48]. This result is consistent with those obtained in the enzymatic activities and tissue amino acid profiles in this study.

## 5. Conclusions

This research showed that dietary supplementation of 0.4–0.8% CPH could increase the growth performance and protein utilization of *E. sinensis*. High levels of CPH (exceeding 1.6%) could enhance amino acid metabolism by activating the IGF-1/Akt/TOR and IGF-1/Akt/FoxO1 pathways, thereby increasing the protein efficiency ratio and protein deposition. Due to the above benefits, CPH has the potential to be used as a growth promoter for other aquatic animals, especially crustaceans. However, the potential mechanisms underlying these benefits of CPH are still not fully elucidated. Further studies are needed to utilize peptidomics technology to screen the functional peptides and validate their functions both in vivo and in vitro. This could further promote the application of cottonseed meal and its by-products in the aquafeed industry. 

## Figures and Tables

**Figure 1 animals-13-03631-f001:**
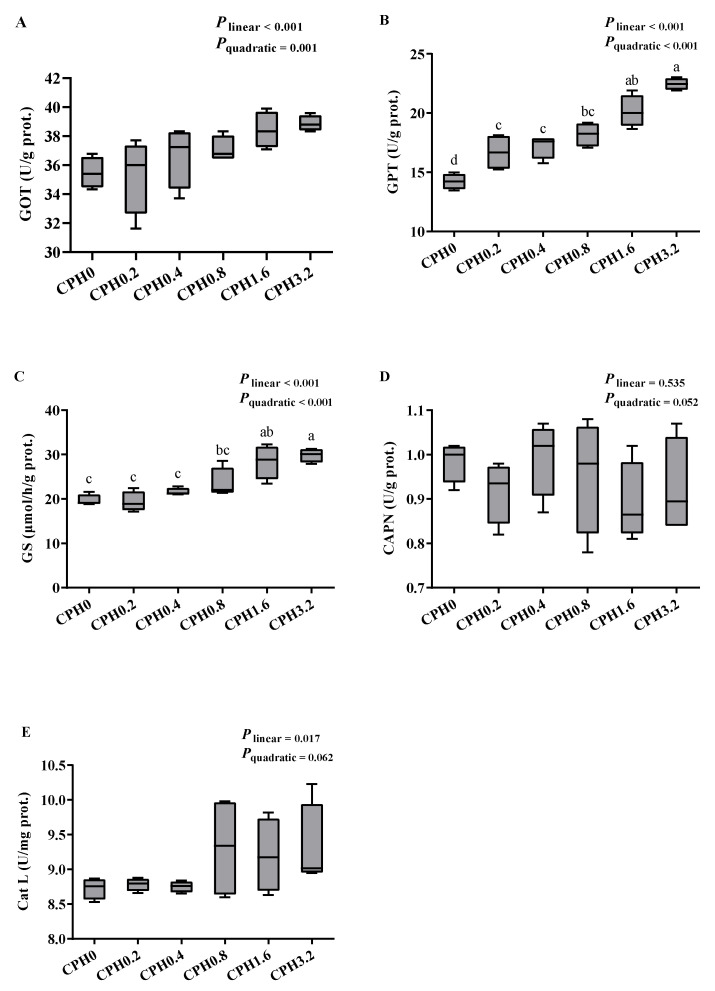
Effects of dietary cottonseed meal protein hydrolysate levels on the activities of enzymes related to protein metabolism in the hepatopancreas of *E. sinensis*. (**A**): aspartate aminotransferase, GOT; (**B**): alanine aminotransferase, GPT; (**C**): glutamine synthetase, GS; (**D**): calpain, CAPN; (**E**): cathepsin L, catL. The upper and lower limits of the box represent the first and third quartiles, while the horizontal line inside the box represents the second quartile (median). Whiskers represent the maximum and minimum values. Each data point represents the mean of four replicates. Boxes assigned different superscripts are significantly different (*p* < 0.05).

**Figure 2 animals-13-03631-f002:**
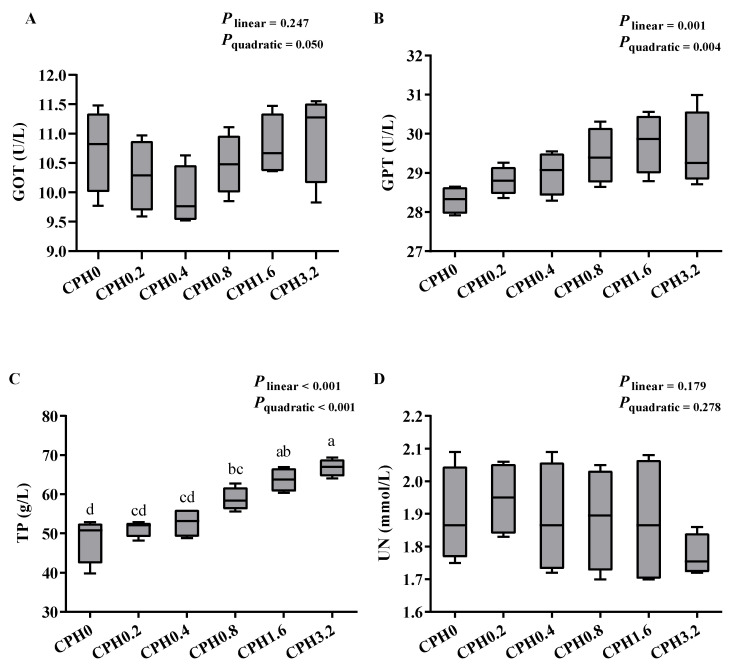
Effects of dietary cottonseed meal protein hydrolysate levels on the activities of enzymes and the concentrations of protein-metabolism-related products in the hemolymph of *E. sinensis*. (**A**): aspartate aminotransferase, GOT; (**B**): alanine aminotransferase, GPT; (**C**): total protein, TP; (**D**): urea nitrogen, UN. The upper and lower limits of the box represent the first and third quartiles, while the horizontal line inside the box represents the second quartile (median). Whiskers represent the maximum and minimum values. Each data point represents the mean of four replicates. Boxes assigned different superscripts are significantly different (*p* < 0.05).

**Figure 3 animals-13-03631-f003:**
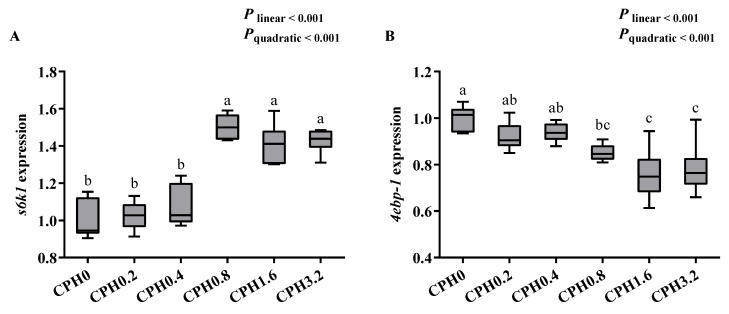
Effects of dietary cottonseed meal protein hydrolysate levels on the relative expressions of S6 kinase-poly-peptide 1 (S6K1, (**A**)) and 4E blinding protein 1 (4EBP-1, (**B**)) in the hepatopancreas of *E. sinensis*. The upper and lower limits of the box represent the first and third quartiles, while the horizontal line inside the box represents the second quartile (median). Whiskers represent the maximum and minimum values. For tissue expression, data are referred to the values (relative units, RU) found in the CPH0 group. Each data point represents the mean of four replicates. Boxes assigned different superscripts are significantly different (*p* < 0.05).

**Figure 4 animals-13-03631-f004:**
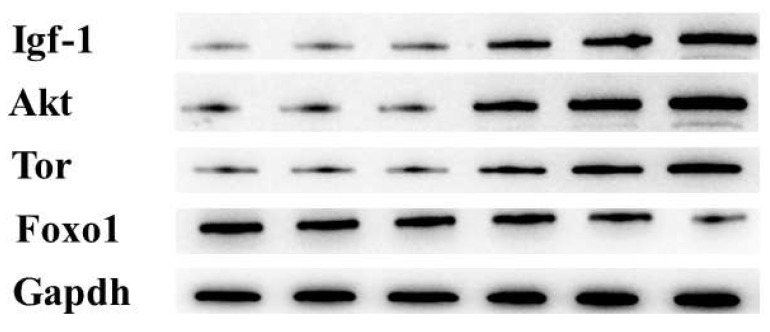
Effects of dietary cottonseed meal protein hydrolysate levels on the protein levels of insulin-like growth factor-1 (Igf-1, (**A**)), protein kinase B (Akt, (**B**)), target of rapamycin, (Tor, (**C**)), and forkhead box O1 (Foxo1, (**D**)) in the hepatopancreas of *E. sinensis.* Each data point represents the mean of four replicates. Boxes assigned different superscripts are significantly different (*p* < 0.05); original Western blot figures are in Appendix A.

**Table 1 animals-13-03631-t001:** Formulation and proximate composition of the experimental diets.

Ingredients (%)	CPH0	CPH0.2	CPH0.4	CPH0.8	CPH1.6	CPH3.2
Fish meal	30.00	30.00	30.00	30.00	30.00	30.00
Soybean meal (defatted)	10.00	10.00	10.00	10. 00	10.00	10.00
Blood meal	4.00	4.00	4.00	4.00	4.00	4.00
Cotton meal	4.00	3.78	3.56	3.12	2.24	0.47
CPH	0.00	0.20	0.40	0.80	1.60	3.20
Peanut meal	18.81	18.81	18.81	18.81	18.81	18.81
Rapeseed meal	2.00	2.00	2.00	2.00	2.00	2.00
α-starch	20.93	20.95	20.97	21.01	21.10	21.26
Soybean oil	3.55	3.55	3.55	3.55	3.55	3.55
Fish oil	1.00	1.00	1.00	1.00	1.00	1.00
Ca(H_2_PO_4_)_2_	1.50	1.50	1.50	1.50	1.50	1.50
Zeolite powder	0.90	0.90	0.90	0.90	0.90	0.90
Premix ^1^	1.00	1.00	1.00	1.00	1.00	1.00
Mixture ^2^	2.30	2.30	2.30	2.30	2.30	2.30
Proximate composition (%, dry-matter basis)						
Dry matter	89.46	89.55	89.36	89.94	89.79	90.36
Crude protein	39.78	39.85	39.91	39.87	39.79	39.78
Ether extract	7.51	7.49	7.48	7.50	7.49	7.46
Crude ash	11.78	11.89	11.95	11.82	12.01	11.91
Crude fiber	4.42	4.57	4.45	4.38	4.51	4.48

^1^ Premix supplied the following minerals (g/kg) and vitamins (IU or mg/kg) per kg: CuSO_4_·5H_2_O, 2.0 g; FeSO_4_·7H_2_O, 25 g; ZnSO_4_·7H_2_O, 22 g; MnSO_4_·4H_2_O, 7 g; Na_2_SeO_3_, 0.04 g; KI, 0.026 g; CoCl_2_·6H_2_O, 0.1 g; Vitamin A, 900,000 IU; Vitamin D, 200,000 IU; Vitamin E, 4500 mg; Vitamin K_3_, 220 mg; Vitamin B_1_, 320 mg; Vitamin B_2_, 1090 mg; Vitamin B_5_, 2000 mg; Vitamin B_6_, 500 mg; Vitamin B_12_, 1.6 mg; Vitamin C, 10,000 mg; pantothenate, 1000 mg; folic acid, 165 mg; choline, 60,000 mg; biotin, 100 mg; myoinositol 15,000 mg; ^2^ Mixture includes the following ingredients (%): choline chloride 4.21%; antioxidants 1.26%; mildew-proof agent 2.09%; salt 21.03%; Lvkangyuan 63.15%; and biostimep 8.26%. The crude protein, ether extract, crude ash, and gross energy contents of cottonseed meal and cottonseed meal protein hydrolysate (CPH) on a dry matter basis were 54.00% vs. 59.54%, 0.74% vs. 0.97%, 6.45% vs. 6.87%, and 11.1 MJ/kg vs. 12.3 MJ/kg, respectively. The crude proteins of fishmeal, soybean meal, blood meal, and peanut meal on a dry matter basis were 64.34%, 45.79%, 90.02%, and 50.85%, respectively. CPH0, dietary cotton meal replacement by 0% CPH; CPH0.2, dietary cotton meal replacement by 0.2% CPH; CPH0.4, dietary cotton meal replacement by 0.4% CPH; CPH0.8, dietary cotton meal replacement by 0.8% CPH; CPH1.6, dietary cotton meal replacement by 1.6% CPH; and CPH3.2, dietary cotton meal replacement by 3.2% CPH.

**Table 2 animals-13-03631-t002:** Amino acid composition of the experimental diets (g/kg, air-dry basis).

Amino Acids	CPH0	CPH0.2	CPH0.4	CPH0.8	CPH1.6	CPH3.2
Essential amino acids						
Threonine	15.4	15.3	15.4	15.4	15.4	15.3
Valine	19.2	19.2	19.3	19.2	19.5	19.1
Methionine	7.7	7.7	7.8	7.6	7.7	7.8
Isoleucine	14.5	14.6	14.8	14.5	14.4	14.7
Leucine	30.0	29.2	29.3	29.7	29.4	29.5
Phenylalanine	18.5	18.8	18.6	18.7	19.1	18.5
Lysine	24.8	24.8	24.6	24.5	24.8	25.1
Histidine	10.8	10.7	10.8	10.9	11.1	11.3
Arginine	22.3	22.2	22.8	23.5	23.3	25.4
Σ Essential amino acids	163.2	162.5	163.4	164.0	164.7	166.7
Non-essential amino acids						
Asparagine	33.1	32.9	32.3	33.2	33.1	33.4
Serine	17.8	17.6	17.5	17.1	17.6	17.7
Glutamic acid	46.5	47.7	47.2	46.9	47.8	47.1
Glycine	21.8	21.4	21.1	20.9	21.8	21.7
Alanine	21.2	21.5	21.3	21.3	21.5	21.8
Cysteine	4.4	4.3	4.1	4.2	4.4	4.3
Tyrosine	13.6	13.4	13.0	13.4	13.5	13.6
Proline	16.4	15.9	16.2	16.3	16.1	16.5
Σ Non-essential amino acids	174.8	174.7	172.7	173.3	175.8	176.1

**Table 3 animals-13-03631-t003:** Nucleotide sequences of the primers used in real-time PCR.

Gene	Forward (5′-3′)	Reverse (5′-3′)	Amplicon Size (bp)	Reference
*s6k1*	TCAATAGCGTCGTCATCG	CCCTGCGTGTAGTGGTTG	164	[21]
*4ebp1*	GCAACACGCCAACTAAACTC	GCGACACCACCTAATATCCA	352	[21]
*s27*	GGTCGATGACAATGGCAAGA	CCACAGTACTGGCGGTCAAA	105	[22]
*β-actin*	TCGTGCGAGACATCAAGGAAA	AGGAAGGAAGGCTGGAAGAGTG	178	KM244725.1

*s6k1*, S6 kinase-polypeptide 1; *4ebp1*, 4E-binding protein 1; *s27*, ubiquitin/ribosomal S27 fusion protein.

**Table 4 animals-13-03631-t004:** Effects of dietary cottonseed meal protein hydrolysate levels on the growth performance of *E. sinensis*.

Group	CPH0	CPH0.2	CPH0.4	CPH0.8	CPH1.6	CPH3.2	Regression Analysis	
							Linear	Quadratic
IW (g)	37.35 ± 0.05	37.40 ± 0.16	37.20 ± 0.22	37.25 ± 0.10	37.60 ± 0.52	37.00 ± 0.01	0.503	0.694
SGR (%/day) ^1^	0.86 ± 0.03 ^c^	0.90 ± 0.02 ^bc^	0.91 ± 0.01 ^bc^	0.94 ± 0.01 ^ab^	0.96 ± 0.02 ^ab^	1.01 ± 0.03 ^a^	<0.001	<0.001
ABW (%) ^2^	48.93 ± 0.59 ^c^	49.57 ± 0.3 ^bc^	49.62 ± 0.11 ^bc^	50.25 ± 0.16 ^abc^	50.94 ± 0.19 ^ab^	51.09 ± 0.3 ^a^	<0.001	<0.001
HSI (%) ^3^	4.72 ± 0.07 ^b^	5.06 ± 0.19 ^ab^	5.39 ± 0.12 ^a^	5.52 ± 0.14 ^a^	5.51 ± 0.14 ^a^	5.45 ± 0.13 ^a^	<0.001	<0.001
CF ^4^	0.48 ± 0.01	0.49 ± 0.01	0.49 ± 0.01	0.49 ± 0.04	0.49 ± 0.04	0.49 ± 0.02	0.657	0.529
PER ^5^	1.21 ± 0.06 ^c^	1.27 ± 0.03 ^bc^	1.30 ± 0.01 ^bc^	1.36 ± 0.02 ^abc^	1.39 ± 0.02 ^ab^	1.48 ± 0.03 ^a^	<0.001	<0.001
NRE (%) ^6^	36.78 ± 1.22 ^c^	37.3 ± 0.99 ^c^	38.57 ± 0.18 ^bc^	40.95 ± 0.42 ^ab^	42.11 ± 0.60 ^a^	42.95 ± 0.43 ^a^	<0.001	<0.001
MR (%) ^7^	27.50 ± 7.70	35.00 ± 2.89	22.50 ± 2.50	35.00 ± 5.00	30.00 ± 4.08	32.50 ± 4.79	0.585	0.859

Values are means ± S.E.M. of four replicates. Means in the same line with different superscript letters are significantly different (*p* < 0.05); ^1^ Specific growth rate (SGR, %/day) = (LnW_t_ − LnW_0_) × 100/T; ^2^ Average body weight (ABW, g) = (W_0_ + W_t_)/2; ^3^ Hepatopancreas index (HSI, %) = (hepatopancreas weight (g) × 100)/body weight (g); ^4^ Condition factor (CF) = (body weight (g) × 100)/total body length(cm)^3^; ^5^ Protein efficiency ratio (PER, %) = wet weight gain (g)/total protein fed (g); ^6^ Nitrogen retention efficiency (NRE, %) = [(W_t_ × C_t_) − (W_0_ × C_0_)] × 100/(C_N_ × feed intake); ^7^ Mortality rate (MR, %) = 100 − (N_F_/N_I_) × 100; W_t_ and W_0_ are the final and initial body weights of crab; T is the culture period in days; C_t_ and C_0_ are the final and initial contents in the whole body of crab, respectively; C_N_ is the nitrogen in diets; N_F_ and N_I_ are the final and initial number of crab.

**Table 5 animals-13-03631-t005:** Effects of dietary cottonseed meal protein hydrolysate levels on the whole-body composition (%, wet-weight basis) of *E. sinensis*.

Group	Initial	CPH0	CPH0.2	CPH0.4	CPH0.8	CPH1.6	CPH3.2	Regression Analysis	
								Linear	Quadratic
Moisture (%)	75.76	73.41 ± 0.39 ^a^	73.92 ± 0.32 ^a^	73.45 ± 0.3 ^a^	72.83 ± 0.18 ^ab^	71.54 ± 0.31 ^c^	71.69 ± 0.11 ^bc^	<0.001	<0.001
Crude protein (%)	9.86	11.73 ± 0.17 ^bc^	11.66 ± 0.23 ^c^	12.01 ± 0.08 ^abc^	12.49 ± 0.16 ^ab^	12.64 ± 0.24 ^a^	12.71 ± 0.11 ^a^	<0.001	<0.001
Ether extract (%)	2.79	3.09 ± 0.08	3.29 ± 0.15	3.19 ± 0.14	3.35 ± 0.02	3.39 ± 0.06	3.36 ± 0.14	0.049	0.122

Values are means ± S.E.M. of four replicates. Means in the same line with different superscript letters are significantly different (*p* < 0.05).

**Table 6 animals-13-03631-t006:** Effects of dietary cottonseed meal protein hydrolysate levels on the hemolymph free amino acid composition (nmol/mL) of *E. sinensis*.

Amino Acids	CPH0	CPH0.2	CPH0.4	CPH0.8	CPH1.6	CPH3.2	Regression Analysis	
							Linear	Quadratic
EAA								
Threonine	277.67 ± 2.60 ^b^	281.33 ± 0.88 ^ab^	281.67 ± 3.28 ^ab^	286.00 ± 3.06 ^ab^	292.00 ± 0.88 ^ab^	292.00 ± 3.06 ^a^	<0.001	<0.001
Valine	84.00 ± 1.00 ^b^	87.33 ± 0.88 ^ab^	86.00 ± 1.15 ^ab^	87.33 ± 1.20 ^ab^	87.33 ± 1.33 ^ab^	90.33 ± 2.02 ^a^	0.008	0.031
Methionine	44.00 ± 1.53	42.21 ± 1.15	47.00 ± 1.00	42.33 ± 0.33	42.01 ± 1.00	45.00 ± 2.52	0.981	0.972
Isoleucine	21.33 ± 0.67 ^b^	24.00 ± 0.58 ^ab^	21.67 ± 0.67 ^ab^	24.33 ± 0.67 ^ab^	25.00 ± 0.58 ^a^	24.33 ± 0.41 ^ab^	0.008	0.028
Leucine	25.00 ± 1.73	25.67 ± 1.45	24.67 ± 0.67	25.00 ± 0.58	28.00 ± 1.00	27.33 ± 0.33	0.051	0.103
Phenylalanine	33.00 ± 1.53	34.33 ± 1.86	36.00 ± 0.58	32.00 ± 1.00	36.33 ± 2.19	37.33 ± 1.67	0.103	0.233
Lysine	104.33 ± 0.88 ^b^	104.67 ± 0.33 ^b^	106.67 ± 1.76 ^ab^	109.00 ± 1.00 ^ab^	113.67 ± 2.67 ^a^	113.33 ± 1.76 ^a^	<0.001	<0.001
Histidine	62.67 ± 1.20 ^b^	66.33 ± 1.45 ^ab^	64.67 ± 0.88 ^ab^	69.67 ± 1.20 ^ab^	70.01 ± 0.58 ^a^	71.33 ± 2.67 ^a^	<0.001	0.001
Arginine	415.00 ± 9.54	434.67 ± 15.30	441.33 ± 11.26	437.00 ± 3.06	421.00 ± 6.11	449.00 ± 9.29	0.175	0.363
Σ EAA	1067.00 ± 7.55 ^b^	1100.33 ± 16.33 ^ab^	1109.67 ± 13.86 ^ab^	1112.67 ± 5.46 ^ab^	1111.00 ± 4.36 ^ab^	1150.00 ± 12.50 ^a^	<0.001	0.001
NEAA								
Asparagine	23.67 ± 0.67	24.00 ± 0.58	24.67 ± 0.33	25.00 ± 0.58	24.33 ± 0.33	24.67 ± 0.33	0.126	0.151
Serine	265.00 ± 2.00	267.33 ± 3.38	263.67 ± 3.71	270.00 ± 7.64	276.67 ± 3.84	279.33 ± 0.88	0.005	0.012
Glutamic acid	164.00 ± 1.15	157.33 ± 3.71	158.00 ± 3.46	157.00 ± 6.11	158.67 ± 4.26	162.00 ± 4.04	0.831	0.299
Glycine	632.00 ± 4.00	664.33 ± 18.18	673.67 ± 4.91	660.00 ± 22.54	664.67 ± 16.05	651.00 ± 12.12	0.517	0.141
Alanine	493.67 ± 3.71	498.33 ± 4.70	504.33 ± 3.52	510.33 ± 16.60	522.00 ± 23.50	535.33 ± 2.03	0.006	0.020
Cysteine	44.00 ± 1.15	45.33 ± 4.37	40.67 ± 1.20	50.00 ± 2.65	47.67 ± 3.48	49.33 ± 3.18	0.100	0.249
Tyrosine	46.33 ± 1.33	48.67 ± 1.86	56.00 ± 1.00	51.67 ± 3.28	49.33 ± 2.85	52.67 ± 1.45	0.180	0.142
Σ NEAA	1668.67 ± 5.21	1705.33 ± 12.68	1721.00 ± 13.32	1724.00 ± 19.05	1743.33 ± 43.17	1754.33 ± 11.55	0.003	0.013
Σ AA	2735.67 ± 3.76 ^b^	2805.67 ± 17.67 ^ab^	2830.67 ± 15.39 ^ab^	2836.67 ± 15.45 ^ab^	2854.33 ± 43.85 ^a^	2904.33 ± 14.74 ^a^	<0.001	<0.001

Values are means ± S.E.M. of four replicates. Means in the same line with different superscript letters are significantly different (*p* < 0.05); Σ EAA: Total essential amino acids; Σ NEAA: Total non-essential amino acids; Σ AA: Total amino acids.

**Table 7 animals-13-03631-t007:** Effects of dietary cottonseed meal protein hydrolysate levels on the hepatopancreas amino acid contents (g/kg of fresh weight) of *E. sinensis*.

Amino Acids	CPH0	CPH0.2	CPH0.4	CPH0.8	CPH1.6	CPH3.2	Regression Analysis	
							Linear	Quadratic
EAA								
Threonine	0.40 ± 0.02 ^c^	0.44 ± 0.02 ^bc^	0.44 ± 0.01 ^bc^	0.46 ± 0.01 ^abc^	0.50 ± 0.01 ^ab^	0.51 ± 0.01 ^a^	<0.001	<0.001
Valine	0.41 ± 0.03 ^b^	0.44 ± 0.03 ^ab^	0.44 ± 0.01 ^ab^	0.47 ± 0.01 ^ab^	0.48 ± 0.01 ^ab^	0.51 ± 0.01 ^a^	<0.001	0.002
Methionine	0.20 ± 0.01 ^b^	0.24 ± 0.03 ^ab^	0.25 ± 0.01 ^ab^	0.27 ± 0.03 ^ab^	0.24 ± 0.01 ^ab^	0.32 ± 0.04 ^a^	0.007	0.029
Isoleucine	0.30 ± 0.01 ^b^	0.35 ± 0.03 ^b^	0.35 ± 0.01 ^b^	0.36 ± 0.01 ^b^	0.38 ± 0.01 ^ab^	0.46 ± 0.03 ^a^	<0.001	<0.001
Leucine	0.60 ± 0.06	0.66 ± 0.07	0.66 ± 0.01	0.64 ± 0.02	0.70 ± 0.01	0.80 ± 0.04	0.008	0.015
Phenylalanine	0.37 ± 0.03 ^b^	0.42 ± 0.03 ^ab^	0.41 ± 0.02 ^ab^	0.45 ± 0.01 ^ab^	0.46 ± 0.02 ^ab^	0.51 ± 0.01 ^a^	<0.001	0.001
Lysine	0.55 ± 0.04 ^b^	0.67 ± 0.04 ^b^	0.69 ± 0.04 ^b^	0.67 ± 0.04 ^b^	0.72 ± 0.03 ^ab^	0.93 ± 0.07 ^a^	<0.001	0.001
Histidine	0.19 ± 0.01 ^c^	0.23 ± 0.01 ^bc^	0.22 ± 0.01 ^bc^	0.24 ± 0.01 ^ab^	0.26 ± 0.01 ^ab^	0.28 ± 0.01 ^a^	<0.001	<0.001
Arginine	0.79 ± 0.20	0.64 ± 0.01	0.74 ± 0.05	0.77 ± 0.06	0.75 ± 0.07	1.08 ± 0.21	0.110	0.073
Σ EAA	3.83 ± 0.40 ^b^	4.08 ± 0.26 ^b^	4.18 ± 0.06 ^ab^	4.33 ± 0.12 ^ab^	4.48 ± 0.11 ^ab^	5.39 ± 0.38 ^a^	0.001	0.001
NEAA								
Asparagine	0.73 ± 0.02 ^e^	0.76 ± 0.01 ^de^	0.83 ± 0.01 ^cd^	0.90 ± 0.02 ^bc^	0.96 ± 0.02 ^b^	1.07 ± 0.02 ^a^	<0.001	<0.001
Serine	0.36 ± 0.02 ^c^	0.38 ± 0.02 ^bc^	0.40 ± 0.01 ^abc^	0.42 ± 0.01 ^abc^	0.45 ± 0.02 ^ab^	0.48 ± 0.02 ^a^	<0.001	<0.001
Glutamic acid	0.96 ± 0.10 ^b^	1.03 ± 0.10 ^ab^	1.09 ± 0.05 ^ab^	1.06 ± 0.02 ^ab^	1.14 ± 0.07 ^ab^	1.54 ± 0.21 ^a^	0.004	0.004
Glycine	0.48 ± 0.11	0.53 ± 0.12	0.48 ± 0.01	0.56 ± 0.07	0.49 ± 0.02	0.70 ± 0.09	0.129	0.211
Alanine	0.45 ± 0.06	0.48 ± 0.04	0.49 ± 0.01	0.49 ± 0.022	0.51 ± 0.02	0.65 ± 0.06	0.004	0.005
Cysteine	0.14 ± 0.01 ^bc^	0.15 ± 0.00 ^abc^	0.18 ± 0.01 ^ab^	0.18 ± 0.01 ^a^	0.17 ± 0.01 ^abc^	0.14 ± 0.01 ^c^	0.739	0.001
Tyrosine	0.38 ± 0.01 ^c^	0.39 ± 0.00 ^c^	0.40 ± 0.01 ^bc^	0.40 ± 0.00 ^bc^	0.44 ± 0.01 ^ab^	0.48 ± 0.02 ^a^	<0.001	<0.001
Σ NEAA	3.50 ± 0.30 ^b^	3.71 ± 0.27 ^b^	3.87 ± 0.09 ^b^	4.02 ± 0.09 ^ab^	4.16 ± 0.14 ^ab^	5.05 ± 0.40 ^a^	<0.001	0.001
Σ AA	7.33 ± 0.71 ^b^	7.79 ± 0.53 ^b^	8.06 ± 0.14 ^b^	8.36 ± 0.20 ^ab^	8.64 ± 0.25 ^ab^	10.44 ± 0.78 ^a^	<0.001	0.001

Values are means ± S.E.M. of four replicates. Means in the same line with different superscript letters are significantly different (*p* < 0.05); Σ EAA: Total essential amino acids; Σ NEAA: Total non-essential amino acids; Σ AA: Total amino acids.

## Data Availability

The data of this research can be obtained from the corresponding author upon reasonable request.

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
