# Peer review of "Utilization of Cottonseed Meal Protein Hydrolysate by Crustaceans: Insights on Growth Performance, Protein Turnover, and Metabolism in Chinese Mitten Crab Eriocheir sinensis"

_animals, 2023, doi:10.3390/ani13233631_

Round 1

Reviewer 1 Report

Comments and Suggestions for Authors

Finding alternative protein-based ingredients is always worthy to study and publish in aquaculture industry. The manuscript is well-written and easy to read and understand. The gold point of this study is to evaluate the hydrolysate functions in the protein turnover and metabolism of crab. Therefore, it is highly recommend to publish in Animals. There are some comments that need to be clarified by the authors prior to publication as follows:

-Please change the title by using cottonseed meal protein hydrolysate instead of plant protein hydrolysates. Because plant protein hydrolysate is so general for a research paper and then you use it as one of the keywords.

-L27: It is not necessary to use “ respectively

-According to L33, please indicate the control group in L26.

-L40: what is the final conclusion? recommended dose(s) for the industry? or any recommendation(if needed)

-L45-56: any information at world level?

L64: as far as I know, corn meal hydrolysate is commercialize and practical in aquaculture nutrition.

- Table 1: Since the authors replaced Cotton meal with CPH to balance the experimental diets, please prove the proximate composition (Pro, EE, Ash, and Moisture) of them in the foot note. Besides, it would be more practical to denote at least the Pro level of fishmeal, Soybean meal, Blood meal, Peanut meal.

- Table 1: Why the authors used Rapeseed meal for the crab?

- Table 1: It is crucial to know the ash and fiber contents of the diets.

- Table 2: The total AAs are questionable. It is confusing that for example the crude protein in CPH0 is 75.1 g/kg dry basis but the total AAs of CPH0 is 338 (163.2+174.8). Please justify this big differences. This question is remained for Table 6 according to Table 5.

-L147: any information regarding ammonia? Water systems or flow rate? Aeration?

-Please use h instead of hours/ min instead of minutes

-L158: use ×g (RCF) instead of rpm for the centrifugation methods. Plz check it throughout the manuscript.

-L171: Are they characterized for crustaceans or fish or human?

-L158: What was the features of the internal standard used? I reckon Argnine has another method to measure.

-As total n ≥50, why the authors used Shapiro-Wilk test instead of Kolmogorov–Smirnov test?

- I read the discussion section and it is concise, insightful, and accepted.

-L371: Please remove the parenthesis.

- Conclusion is accepted.

-L447: Any recommendation for future studies?

Author Response

Thank you very much for taking the time to review this manuscript,we have responded to your comments and suggestions point by point and highlighted the changes throughout the article.Please see the attachment.

Reviewer 2 Report

Comments and Suggestions for Authors

Abstract

The authors could discuss the practical implications of their findings in more detail. For example, they could discuss the optimal level of CPH supplementation for E. sinensis and the cost-effectiveness of using CPH as a feed additive.

The authors could also discuss the potential applications of their findings to other aquaculture species.

Introduction

The authors could consider adding a diagram to illustrate the IGF-1/Akt/TOR and IGF-1/Akt/FoxO1 pathways. This would help readers to understand the molecular mechanisms underlying the effects of CPH on protein turnover and metabolism.

The authors could also discuss the potential applications of their findings to the development of improved artificial feed for E. sinensis and other aquatic animals.

·       In the Introduction, the authors could expand on the economic significance of E. sinensis aquaculture and the challenges posed by the government's restriction on the use of trash fish in crab feed. This would provide a stronger context for the study.

In the Methods section, the authors could provide more detail about the CPH diet, such as the source of the cottonseed meal, the degree of hydrolysis, and the amino acid composition.

In the Results section, the authors could present the data in a more concise and informative way. For example, instead of presenting all of the data in tables, the authors could summarize the key findings in the text and then present the tables in the supplementary material.

Discussion

·       The authors could consider adding a diagram to illustrate the IGF-1/Akt/TOR and IGF-1/Akt/FoxO1 pathways in crustaceans. This would help readers to understand the molecular mechanisms underlying the effects of CPH on protein turnover and metabolism.

·       The authors could also discuss the potential applications of their findings to the development of improved artificial feed for E. sinensis and other aquatic animals.

Overall comments

The manuscript is well-written and informative. The authors have done a good job of describing the experimental design and results, and discussing the implications of their findings. The authors have also adequately addressed the comments of the previous reviewers.I recommends its minor revision.

Comments on the Quality of English Language

a through revision for language and phrasing is highly recommended 

Author Response

(The authors gave the same response as above.)

Reviewer 3 Report

Comments and Suggestions for Authors

the work is interesting but the writing needs to be heavily revised, there are many inaccuracies and even errors in the materials and methods, the introduction is very lacking and must be significantly improved if the article is to be published. the results are not very clear but I think it is more of a linguistic issue. I suggest a new draft of the work for a new submission

Line 16: substitute  CPH with cottonseed meal protein hydrolysate ( CPH)

Line 23: the purpose of who?

The abstract is not clear I suggest revising it, A phrase is 6 lines ( 27-33) a bit to long and unclear

Line 45: Chinese mitten crab There is a widespread Incorrect

Line 49: phrase unclear, al the period from 44-56 it’s to short and not well explained

Line 77-93: unclear period where the authors explain the metabolic  pathway.

Line 107: Where were the raw materials purchased from?

Line 111: Producer? AS1.398 neutral protease

Line 143 : unclear, are used 10 craps x pools?

Line 146: treated? Or feed?

Line 161: mention all the instrument producers.

Line 173: which spectrophotometer?

Line 176: unclear protocol

Line 181: unclear protocol, add information

Line 188: Which instrument?

Line 200: Sigma, …. It’s missing the country

Line 207: I’m not sure about this step, please provide additional information

Line 210-214: protocol incorrect ( checked chrome-extension://efaidnbmnnnibpcajpcglclefindmkaj/https://www.takarabio.com/documents/User%20Manual/RR036A_e.v2008Da.pdf).

Line 216: Please use same format 4E-BP1 or 4EBP-1, check in the manuscript

Line 217: which are the  HKG used?justs27? I suggest at least 2.

Line 225 : producer?

Line 231 : instrument?

Line 271: why just 4? Why weren't they all weighed? I believe the use of only 4 biological replicates is insufficient to evaluate the data

Line 339: general question, why do you evaluate just these 2 genes? The gene expression is poor, and

Line 371: typing errors

Line 372: These results indicated that CPH supplemented at appropriate doses might improve the growth performance , or you have a improve or not… but if you evaluate just 4 individuals is not a trustable data

Comments on the Quality of English Language

English needs to be revised, text is not clear

Author Response

(The authors gave the same response as above.)

Round 2

Reviewer 1 Report

Comments and Suggestions for Authors

The authors have carefully considered the comments and tried their best to address every one of them.

Regarding to the internal reference, I meant about the standard used to identify the peaks that were used to measure amino acids in this research. You can add the standard reference specifications in the proof version of the article.

Accepted and good luck

Reviewer 3 Report

Comments and Suggestions for Authors

Dear Author, thanks for your clarifications but still some work must be done.

Comments 18: Line 210-214: protocol incorrect (checked chrome-extension://efaidnbmnnnibpcajpcglclefindmkaj/https://www.takarabio.com/documents/User%20Manual/RR036A_e.v2008Da.pdf).

Response 18: We apologize that we only described the step of cDNA production, and did not describe the subsequent process of real-time quantitative PCR. We have added the relevant information in lines 241-248 in the revised manuscript.

Line 239: if the protocol provided  ( cDNA production) is incorrect you should correct it not eliminate it.

Comments 20: Line 217: which are the HKG used? Just S27? I suggest at least 2.

Response 20: We understand your concern about using only one housekeeping gene. The accuracy of qPCR depends on the stability of the reference gene used for data normalization. A previous study has evaluated the expression stability of 10 candidate reference genes at different developmental stages, tissues, and molting stages of, Eriocheir sinensis. The results showed that s27 was the most stable reference gene (Huang et al., 2017). Therefore, only s27 was selected as the housekeeping gene in this study.

Reference:

Huang, S., Chen, X., Wang, J., Chen, J., Yue, W., Lu, W., Lu, G., Wang, C. (2017). Selection of appropriate reference genes for qPCR in the Chinese mitten crab, Eriocheir sinensis (Decapoda, Varunidae). Crustaceana, 90(3), 275-296. DOI:10.1163/15685403-00003651

The accuracy of qPCR depends on the stability of the reference gene used for data normalization. However, the stably expressed reference genes in the Chinese mitten crab, Eriocheir sinensis, have not been well identified under different experimental conditions. In this study, the stabilities of the expressions of 10 candidate reference genes were evaluated in different developmental stages, tissues, and moulting stages of E. sinensis. Our results indicated that UBE and S27 were the most stable reference genes. To validate the suitability of the reference genes, the EcR (ecdysone receptor) gene was analysed among different moulting stages. The results showed that the expression level of EcR was elevated using the least stable reference gene, GST, compared with using the three most stable reference genes. Taken together, our results indicate that reference genes should be assessed and selected in accordance with the experimental conditions, and more than one reference gene should be selected.

You wrote that “more than one reference gene should be selected”, so if you want to provide a valid manuscript I suggest to implement the article with the missing experiments.

Response 24: TOR is a central regulator of eukaryotic protein synthesis, and the S6 kinase-poly-peptide 1 (S6K1) and 4E blinding protein 1 (4EBP-1) are the two main downstream targets of TOR. Therefore, the transcriptions of both genes were determined to reveal the downstream effects of TOR. In addition, there are only 10 crabs in each pool. And mortality was observed during the 12 week culture period, resulting in a limited number of samples here. This might result in the poor gene expression data. Despite this, significance was still observed between different treatments after SPSS analysis. Therefore, we believe that the gene expression data is reliable to some extent.

You mentioned : “And mortality was observed during the 12 week culture period, resulting in a limited number of samples here”, why I  can’t find any information about that? If the experiments were compromised for any diseases or incorrect procedure must be done again, in contrary the data obtained are not trustable. 

I noticed the work done on the manuscript following the requested indications, but important corrections still need to be made and a laboratory part must be performed to make the work acceptable. I also ask you to better explain the issue of mortality that occurred.

Comments on the Quality of English Language

It's improved 
